# Explainable $K$-means Neural Networks for Multi-view Clustering

**Yalan Qin, Xinpeng Zhang, Guorui Feng** [*]
School of Communication and Information Engineering, Shanghai University
{ylqin,xzhang, grfeng}@shu.edu.edu

## Abstract

Despite multi-view clustering has achieved great progress in past decades, it is still a challenge to balance the effectiveness, efficiency, completeness and consistency of nonlinearly separable clustering for the data from different views. To address this challenge, we show that multi-view clustering can be regarded as a three-level optimization problem. To be specific, we divide the multi-view clustering into three sub-problems based on $K$-means or kernel $K$-means, i.e., linear clustering on the original multi-view dataset, nonlinear clustering on the set of obtained linear clusters and multi-view clustering by integrating partition matrices from different views obtained by linear and nonlinear clustering based on reconstruction. We propose Explainable $K$-means Neural Networks (EKNN) and present how to unify these three sub-problems into a framework based on EKNN. It is able to simultaneously consider the effectiveness, efficiency, completeness and consistency for the nonlinearly multi-view clustering and can be optimized by an iterative algorithm. EKNN is explainable since the effect of each layer is known. To the best of our knowledge, this is the first attempt to balance the effectiveness, efficiency, completeness and consistency by dividing the multi-view clustering into three different sub-problems. Extensive experimental results demonstrate the effectiveness and efficiency of EKNN compared with other methods for multi-view clustering on different datasets in terms of different metrics.

## 1 Introduction

As an important problem in machine learning and data mining, clustering aims to group a set of data points into clusters Qin et al. (2021; 2022b; 2023c;a; 2024a; 2023b; 2025f); Pu et al. (2023); Qin et al. (2025d;g;b;h); Qin & Qian (2024); Qin et al. (2024b;c; 2025e;a;c; 2023d); Li et al. (2023a;b); Liu et al. (2024; 2023a; 2025); Lu et al. (2024); Zhang et al. (2025); Li et al. (2025); Liu et al. (2023b; 2022b;a). The data points in different clusters are highly dissimilar but remarkably similar with data points in the same cluster. It has different applications in many fields including image processing Sandler & Lindenbaum (2011), bioinformatics Perner (2002) and signal processing Gupta & Xiao (2011). Various clustering algorithms have been presented Veldt et al. (2019); Gebru et al. (2016), which can be divided into four categories including hierachical, partitional, grid-based and density-based clustering Camastra & Verri (2005); Ng et al. (2001). However, some representative methods Fukunaga & Hostetler (1975); Mehmood et al. (2015) belonging to these categories need to calculate the distances between all data points in the dataset. Consequently, it needs an extremely high computational cost when dealing with large-scale datasets.

Many methods have been developed to improve the efficiency of clustering algorithms by reducing unnecessary calculations of distances between data points Bentley (1975); Viswanath & Pinkesh (2006); He et al. (2011), e.g., spatial index structure Bentley (1975), hybrid clustering Viswanath & Pinkesh (2006), grid-based clustering and parallel clustering He et al. (2011). However, most existing methods pay more attention to the effectiveness and the improvements in efficiency is not so significant as effectiveness. Recently, some methods have been proposed to directly improve the clustering speed of the existing work Mehmood et al. (2015) without considering extra conditions Wu & Wilamowski (2017); Zhang et al. (2016). These methods fail to fully explore how to reduce

---

[*]Corresponding author

the calculations of distances between data points. As a well-known algorithm, $K$-means Queen (1966) belongs to the linearly separable clustering algorithm and has a low computational cost. It is not able to well recognize clusters which are nonlinearly separable, leading to unsatisfied clustering performance.

Nonlinear clustering is a widely exploited problem due to the complex structure of the data in the real world. Recently, various types of nonlinear clustering methods have been proposed Ng et al. (2001); Schölkopf et al. (1998); Cheng (1995) to better handle clusters with arbitrary shapes. Methods based on kernel extract nonlinear separable clusters by adopting a proper nonlinear mapping between the input space to a feature space with high dimensions. This nonlinear mapping is called the kernel function and the representative methods are spectral clustering Ng et al. (2001), kernel $K$-means Schölkopf et al. (1998) and mean-shift Cheng (1995). To determine the relation of each data point to the clusters, these methods need to compute the pairwise similarities of data points. The reason why they are able to recognize nonlinear clusters is that these approaches employ all data points to represent a cluster without choosing a center. However, their space or time costs are high, resulting in unsatisfied results for large-scale datasets. To effectively denote a nonlinear cluster, methods based on multi-clusters have been proposed Liu et al. (2009); Liang et al. (2012); Wang et al. (2013). These methods are able to reduce the computation complexity results by replacing data points with different centers. Therefore, their clustering performances are easily influenced by the chosen center and they also need high computational costs to obtain centers with high quality.

Information is usually presented in different forms simultaneously in the real world, including multiple types of features and multiple modalities Sui et al. (2018); Li & Tang (2017). We consider different types of information as multiple views describing objects. Given web pages, we can extract different types of features based on hyperlinks, text and possible visual information. Multiple types of features can be extracted based on text, edge and color for images. Various existing methods Zhang et al. (2020b;a) have demonstrated that obvious performance improvement can be achieved since different views complete each other. In multi-view learning, consistency and completeness are two well-known principles. To achieve the consistency of different views, the existing work assume that correlations among different views should be maximized Wang et al. (2015). The complete information is important to obtain comprehensive representations. However, existing work fails to balance the effectiveness, efficiency, consistency and completeness of multi-view clustering. Accordingly, a challenging problem appears - how to effectively and efficiently perform nonlinear clustering for multi-view datasets.

Motivated by the fact that different clusters in a complex dataset are nonlinearly separable in the global geometric space but linearly separable in the local geometric space, we assume that some small linearly separable clusters make up a nonlinearly separable cluster. Different from existing studies, we show that multi-view clustering can be treated as a three-level optimization problem based on this assumption, which is presented in Fig. 1. Our method is expected to have the following merits: effectiveness - the quality of multi-view clustering results is guaranteed, efficiency - the computational cost is able to be lowered, completeness and consistency - the clustering results incorporate information from different views and multiple views are consistent, where the symbol "-" is used for explanation. We divide the multi-view clustering into three different sub-problems: 1) the linear clustering on the set of original data points for every view, 2) the nonlinear clustering of each view on the set of linear clusters obtained by 1), and 3) multi-view clustering by incorporating partition matrices from different views obtained by linear and nonlinear clustering based on reconstruction. Here, the linear clustering makes data points similar with each other in the local geometric space into the same clusters, which reduces the size of the data for nonlinear clustering. Then, the efficiency is considered in this manner. Next, we perform nonlinear clustering to partition the linear clusters into different nonlinear clusters. In detail, we aim to project a given multi-view dataset into a feature space where the effectiveness, efficiency, completeness and consistency are guaranteed using three jointly learning objective functions. These three objective functions consist of linear clustering, nonlinear clustering and multi-view clustering, which are all defined based on the $K$-means or kernel $K$-means objective. As convolutional operation in Convolutional Neural Networks (CNN), $K$-means plays the same role in $K$-means Neural networks. The connections of different $K$-means components in Fig. 1 have the same usage as neural network in CNN. Therefore, we propose Explainable $K$-means Neural Networks (EKNN) based on $K$-means for multi-view clustering and use the iterative method in $K$-means to optimize the networks. Besides, it is explainable since the effect of each layer in EKNN is knowable, i.e., the layer of kernel $K$-means is adopted to obtain nonlinear

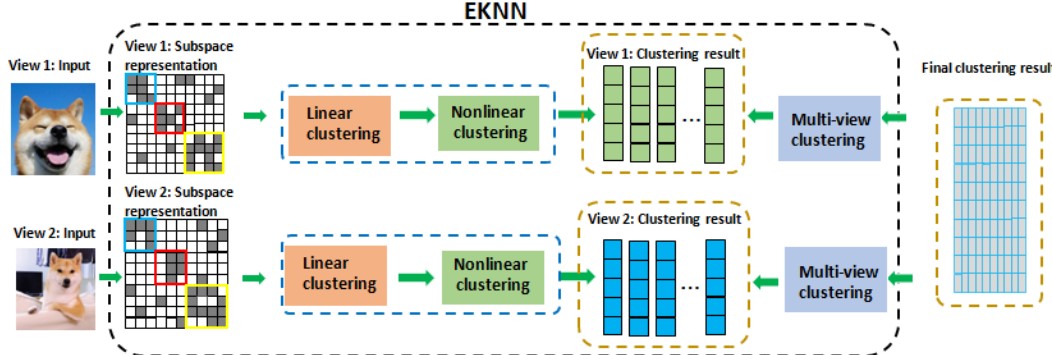

Figure 1: Framework of EKNN. Given the multi-view data as input, subspace representations from different views can be obtained based on self-representative coefficient matrix. The layer of linear clustering, nonlinear clustering and multi-view clustering are used to achieve the goal of efficiency, effectiveness, consistency and completeness, respectively. After linear and nonlinear clustering, we can obtain clustering result for single view. With EKNN, final clustering result can be achieved for the multi-view data.

clusters. We can also observe that $K$-means and kernel $K$-means are special cases of EKNN. We also extend EKNN for multi-view clustering to the case of multi-view subspace learning, which is able to learn a desired latent representation shared by different views. The shared subspace representation can be obtained by the latent representation based on self-expressiveness Elhamifar & Vidal (2013), which is used to achieve the final results by the existing clustering algorithms, i.e., spectral clustering algorithm.

The main contributions in this work are:

- We give a novel insight to the multi-view clustering community, i.e., the multi-view clustering can be regarded as a three-level optimization problem. We propose to divide the multi-view clustering into three sub-problems: the linear clustering on the set of original data points based on $K$-means, the nonlinear clustering of each view on the set of linear clusters based on kernel $K$-means, and multi-view clustering by integrating information from different views for the set of partitions in terms of $K$-means.

- We propose Explainable $K$-means Neural Networks (EKNN) based on $K$-means, which flexibly integrates the linear clustering, nonlinear clustering and multi-view clustering into a framework. It is explainable and able to balance the clustering quality, computation cost, completeness and consistency of multi-view clustering. The iterative method as in $K$-means is used to solve the optimization problem of EKNN for multi-view clustering.

- We further extend EKNN for multi-view clustering to the case of multi-view subspace learning, which is able to learn a desired latent representation shared by different views. Extensive experiments performed on different datasets demonstrate the efficiency and effectiveness of EKNN for multi-view clustering in terms of different metrics compared with the existing clustering algorithms.

The rest of this paper is organized as follows. Section 2 reviews some related work of $K$-means algorithm and multi-view representation learning. Section 3 describes the details of EKNN and the extension of EKNN. In Sections 4 and 5, we conduct experiments to validate the merits of EKNN and show the conclusion, respectively.

## 2 THE PROPOSED METHOD

In this section, we present that multi-view clustering can be seen as a three-level optimization problem and propose EKNN based on $K$-means for multi-view clustering. For multi-view dataset, we first explore the properties of each single view and then integrate them to realize multi-view clustering. For each view of a complex dataset, different clusters are nonlinearly separable in the global

geometric space but linearly separable in the local geometric space. Therefore, for each view, we assume that a linearly separable cluster is made up of the original data points in the dataset and several linearly separable clusters make up a nonlinearly separable cluster. Based on this assumption, the final shared clustering result is achieved by integrating partition matrices from different views obtained by linear and nonlinear clustering based on reconstruction, which is complete. The consistency among multiple views can be indirectly obtained by enforcing the final clustering result to be shared by different views. Built on these observations, we divide the multi-view clustering into three sub-problems based on $K$-means or kernel $K$-means. The proposed EKNN unifies these three sub-problems into a framework and we solve it by the iterative optimization. EKNN is explainable and consists of some layers including linear clustering based on $K$-means, nonlinear clustering based on kernel $K$-means and multi-view clustering based on $K$-means.

## 2.1 FORMULATED PROBLEM

Given the dataset $X = \{X^1, X^2, ..., X^V\}$, where $X^v \in R^{n \times m}$, $V$ is the number of views, $n$ and $m$ denote the number of total points and dimensions in the dataset, respectively. As abovementioned, the proposed EKNN consists of three parts: 1) linear clustering on the set of data points in the dataset based on $K$-means, 2) nonlinear clustering on the set of linear clusters obtained by 1) based on kernel $K$-means, and 3) multi-view clustering by incorporating partition matrices from different views obtained by linear and nonlinear clustering based on $K$-means. For each view, we first define the part of EKNN for linear clustering based on $K$-means as:

$$f'_{L^v} = ||X^v - W^v V^v||_F^2, \tag{1}$$

where $W^v = [w_{ij}^v] \in R^{n \times p}$ is the partition matrix of data points, $w_{ij}^v$ denotes the membership of the $i$-th data point to the $j$-th linear cluster, $p$ denotes the number of linear clusters in $W$ and $V^v \in R^{p \times m}$ is the cluster matrix of the $v$-th view. Note that the value of $p$ is required to be larger than the number of real clusters $k$ in the dataset. Then data points in the same linear clusters are close to each other in the original space of $X$ when the value of $f_{L^v}$ is low.

Rather than integrating different views in the level of raw features as LMSC Zhang et al. (2017), we employ the subspace representation $\Theta^v = [\theta_{ij}^v] \in R^{n \times n}$ for the $v$-th view based on the self-representative coefficient matrix, which is obtained by subspace clustering in terms of self-representation. Since the global or local space is built on the relative relationship between data points in the extracted feature representation of $X^v$ for the $v$-th view, a linearly separable cluster is made up of the refined data points in the dataset and several linearly separable clusters make up a nonlinearly separable cluster. The subspace representation $\Theta^v$ can reveal the underlying cluster structures of data. Then Eq. (1) is reformulated as:

$$f_{L^v} = ||\Theta^v - W^v V^v||_F^2, \ s.t. \ X^v = X^v \Theta^v, \tag{2}$$

where the dimensions of $W$ and $V$ are $n \times p$ and $p \times n$ here, respectively. The motivation that some clustering methods need to compute pairwise similarity between points is originated from the perspective of the final unified learning framework. The computation of the matrix $\Theta$ with $n \times n$ dimension is the pre-processing step, which is used as the input to the final unified learning framework. Considering the case that $diag(\Theta^v) = 0$ can be removed based on the analysis in Lu et al. (2012), we do not impose the constraint of $diag(\Theta^v) = 0$ to Eq. (2). It is observed that the part of EKNN for linear clustering consists of two layers, i.e., one layer is used for $X^v = X^v \Theta^v$ and the other is employed to achieve $f_{L^v} = ||\Theta^v - W^v V^v||_F^2$. Eq. (2) is the first part of EKNN for linear clustering and we then focus on another two parts. For each view, the second part of EKNN for nonlinear clustering is defined based on kernel $K$-means:

$$\begin{aligned} f_{N^v} &= ||\Phi(V^v) - U^v Z^v||_F^2 \\ &= Tr(K^v) - Tr((\hat{U}^v)^T K^v \hat{U}^v), \end{aligned} \tag{3}$$

where $U^v = [u_{ij}^v] \in R^{p \times k}$ is the partition matrix of $V^v$, $K^v \in R^{p \times p}$ denotes the kernel matrix of $V^v$, $\hat{U}^v = U^v(D^v)^{-\frac{1}{2}}$ is the normalized matrix of $U^v$, $D^v = [d_j^v] \in R^{k \times k}$ with $d_j^v = \sum_{i=1}^{p} u_{ij}^v$, $Z^v$ and $\Phi(V^v)$ are cluster centers and the representations of $V^v$ by a kernel function in the nonlinearly embedded space, respectively. The kernel used in Eq. (3) is Gaussian kernel. Based on Eq. (3),

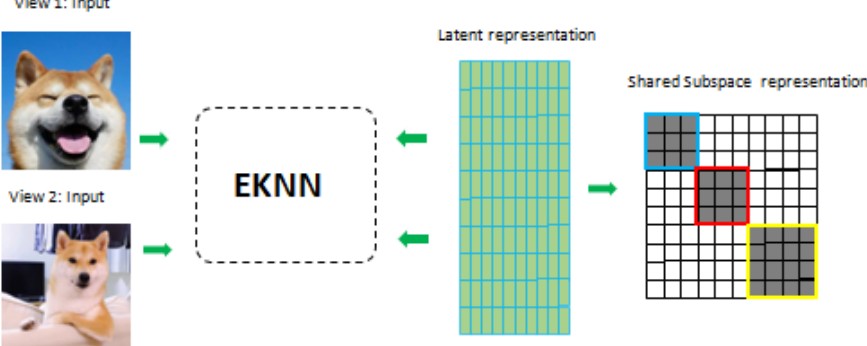

Figure 2: Framework of EKNN for multi-view subspace learning.

the difference of linear clusters belonging to the same nonlinear cluster can be minimized. We then define the last part of EKNN for multi-view clustering based on $K$-means as:

$$
\begin{aligned}
f_{M^v} &= ||W^v \hat{U}^v - \Psi^v(H)||_F^2 \\
&= ||W^v \hat{U}^v - HG^v||_F^2,
\end{aligned}
\tag{4}
$$

where $W^v$ and $\hat{U}^v$ come from Eq. (2) and Eq. (3), respectively. $H \in R^{n \times k}$ denotes the shared partition matrix of the final clustering, $\Psi^v(.)$ is the mapping for the $v$-th view based on reconstruction, $G^v \in R^{k \times k}$ is the relation matrix between $H$ and $W^v \hat{U}^v$. According to the physical meaning of different notations and their dimensions, we combine $W^v$ and $\hat{U}^v$ together in Eq (4). Here, we just consider the simple yet effective mapping, i.e., linear mapping with $G^v$. We can observe that $f_{M^v}$ is the classical $K$-means algorithm when $W^v \hat{U}^v$ is fixed. Based on reconstruction, properties from different views can be encoded into the shared partition matrix $H$ in this manner.

Based on Eqs. (2)-(4), we define the final objective function $F$ of EKNN as:

$$
\min_H F = \sum_{v=1}^{V} \alpha f_{L^v} + \beta f_{N^v} + \gamma f_{M^v}, \ s.t. \ \in \{0,1\}, \ 1 < \sum_{i=1}^{n} w_{ij}^v < n, \ X^v = X^v \Theta^v
$$

$$
\sum_{j=1}^{k} u_{ij}^v = 1, \ 1 < \sum_{i=1}^{p} u_{ij}^v < p, \ w_{ij}^v, h_{ij}^v, u_{ij}^v \in \{0,1\}, \ \sum_{j=1}^{k} h_{ij}^v = 1, \ 1 < \sum_{i=1}^{n} h_{ij}^v < n,
\tag{5}
$$

where $\alpha, \beta, \gamma > 0$ are parameters for balancing different terms. To alleviate the effect of noise, we apply the Frobenius norm for reconstruction loss based on $X^v = X^v \Theta^v$ and use the nuclear norm as regularization term to ensure the high homogeneity within class. We then rewrite Eq. (5) as:

$$
\min_H F = \sum_{v=1}^{V} \alpha ||\Theta^v - W^v V^v||_F^2 + \beta Tr(K^v) - \beta Tr((\hat{U}^v)^T K^v \hat{U}^v) + \gamma ||W^v \hat{U}^v - HG^v||_F^2
$$

$$
+ \eta ||X^v - X^v \Theta^v||_F^2 + \mu ||\Theta^v||_*, \ s.t. \ w_{ij}^v \in \{0,1\}, \ \sum_{j=1}^{p} w_{ij}^v = 1, \ 1 < \sum_{i=1}^{n} w_{ij}^v < n,
$$

$$
u_{ij}^v \in \{0,1\}, \ \sum_{j=1}^{k} u_{ij}^v = 1, \ 1 < \sum_{i=1}^{p} u_{ij}^v < p, \ h_{ij}^v \in \{0,1\}, \ \sum_{j=1}^{k} h_{ij}^v = 1, \ 1 < \sum_{i=1}^{n} h_{ij}^v < n,
\tag{6}
$$

where $\eta, \mu > 0$ are weight parameters. The solution for the above objective function, the corresponding algorithm and complexity analysis are shown in Appendix.

## 2.2 EXPLAINABILITY OF EKNN

Despite the remarkable progress has been achieved by eXplainable Artificial Intelligence (XAI), the explainability usually refers to the understandability of a model built on post-hoc explanations by

different methods, i.e., visual explanations, text explanations, and feature relevance explanations. In this work, we show the explainability from the perspective of the model design, which is more desired in real applications. We express the explainability of the model as transparency, which consists of algorithmic transparency and model decomposability. We then show why EKNN enjoys these two explainable properties in the following.

EKNN owns model decomposability since the input, learned variables and the loss function are all explainable. To be specific, the input to EKNN corresponds to the original dataset, the learned variables have their clear physical meanings and the loss function is built on the $K$-means and kernel $K$-means. EKNN also embraces algorithmic transparency and the dynamic behavior or error surface can be mathematically reasoned, which can make the user better know how the model works.

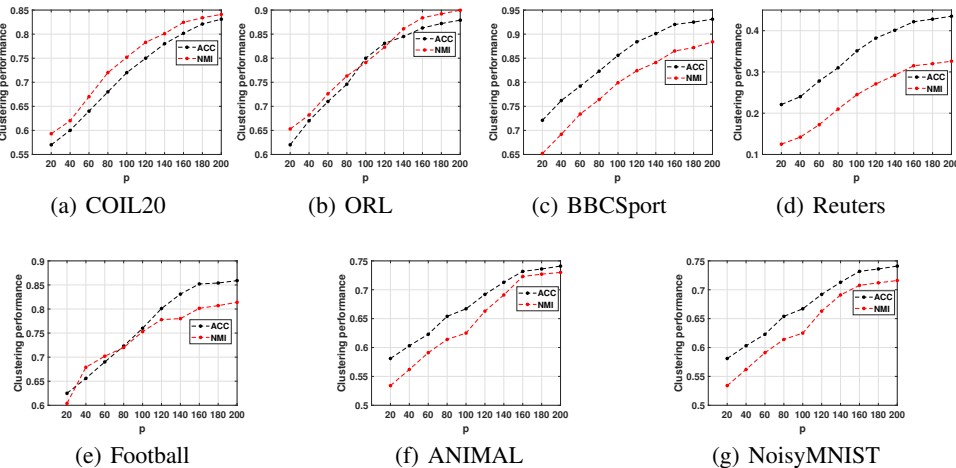

Figure 3: Clustering performance of EKNN with different $p$ on different datasets.

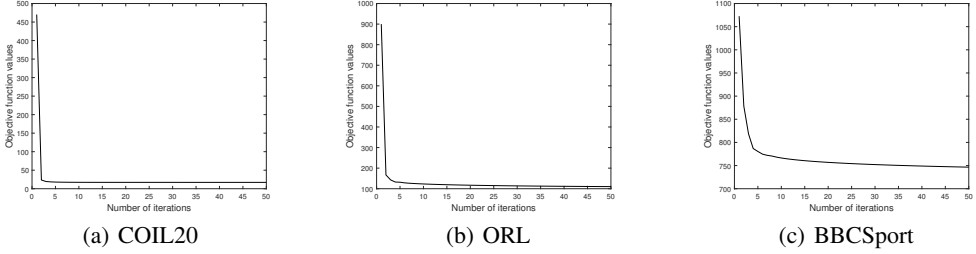

Figure 4: Convergence curve of EKNN on COIL20, ORL and BBCSport dataset.

## 2.3 EXTENSION OF THE PROPOSED METHOD

To learn a more effective latent representation shared by different views for multi-view clustering, we extend EKNN for multi-view clustering to the case of multi-view subspace learning, which is shown in Fig. 2. Note that we introduce shared subspace representation $\Theta_C$ based on the latent representation $H$ for existing clustering algorithms, i.e., spectral clustering algorithm.

Then the final objective function $F$ of EKNN for multi-view subspace learning is formulated as:

$$\min_{H} F = \sum_{v=1}^{V} \alpha ||\Theta^v - W^v V^v||_F^2 + \beta Tr(K^v) + \mu^{'} ||R^{'}||_* - \beta Tr((\hat{U}^v)^T K^v \hat{U}^v) + \eta^{'} ||H - H\Theta_C||_F^2$$

$$+ \eta ||X^v - X^v \Theta^v||_F^2 + \mu ||R^v||_* + \gamma ||W^v \hat{U}^v - HG^v||_F^2,$$

$$s.t.\ w_{ij}^v \in \{0,1\},\ \sum_{j=1}^{p} w_{ij}^v = 1,\ 1 < \sum_{i=1}^{n} w_{ij}^v < n,\ u_{ij}^v \in \{0,1\},\ \sum_{j=1}^{k} u_{ij}^v = 1,\ 1 < \sum_{i=1}^{p} u_{ij}^v < p,$$

$$h_{ij}^v \in \{0,1\},\ \sum_{j=1}^{k} h_{ij}^v = 1,\ 1 < \sum_{i=1}^{n} h_{ij}^v < n,\ \Theta^v = R^v,$$

(7)

where $k_1 > k$, $k_2 > k$, $\mu^{'} > 0, \eta^{'} > 0$ are parameters. Likewise, we can obtain the solution to update $U^v$, $W^v$, $\Theta^v$, $V^v$, $H$ and $G^v$ by employing Algorithm 1. As in $\Theta^v$-sub-problem, we also introduce the auxiliary variable $R^{'}$ and Lagrange multiplier $Y^{'}$ for solving $\Theta_C$. We define $\Phi(Y^{'}, \Theta_C - R^{'}) = \frac{\rho}{2} ||\Theta_C - R^{'}||_F^2 + \langle Y^{'}, \Theta_C - R^{'} \rangle$. For $\Theta_C$-sub-problem, $\Theta_C$, $R^{'}$ and $Y^{'}$ are updated by:

$$\Theta_C = \left[ 2\eta^{'} H^T H + \rho I \right]^{-1} \cdot \left[ 2\eta^{'} H^T H^v + \rho R^{'} - Y^{'} \right],$$

(8)

$$R^{'} = \arg\min_{R^{'}} \frac{\mu^{'}}{\rho} ||R^{'}||_* + \frac{1}{2} ||R^{'} - (\Theta_C + \frac{Y^{'}}{\rho})||_F^2,$$

(9)

$$Y^{'} = Y^{'} + \rho(\Theta_C - R^{'}).$$

(10)

Based on the learned shared subspace representation $\Theta_C$, we use the spectral clustering algorithm to obtain the final clustering results.

## 3 EXPERIMENTAL RESULTS AND ANALYSIS

In this section, we conduct experiments to demonstrate the effectiveness of the proposed EKNN and its extension by comparing the clustering performances of them with some representative subspace clustering approaches on different benchmark datasets. We show the details of the used datasets and compared methods in Appdendix. The parameter selection regarding the parameters in the proposed EKNN is also presented in this Appdendix.

Table 1: Clustering performance (ACC%±STD%) on different datasets.

| Data sets | COMIC | DMF | MVEC | BMVC | DiMSC | RMSL | SSSL-M | Ours | Ours(extension) |
|---|---|---|---|---|---|---|---|---|---|
| COIL20 | 74.23± 0.12 | 72.97± 0.20 | 70.45±2.24 | 73.33± 0.00 | 72.78± 1.44 | 82.19±1.39 | 84.36±0.05 | 80.17±0.18 | 85.22±2.50 |
| ORL | 81.29± 1.00 | 74.45± 2.29 | 71.06± 2.02 | 72.75± 0.00 | 83.84±1.16 | 88.10±1.27 | 91.56±0.01 | 86.28±1.50 | 91.73±2.75 |
| BBCSprot | 95.32 ± 0.18 | 76.84± 0.00 | 96.88± 0.00 | 80.70± 0.00 | 95.10± 2.17 | 97.61±0.18 | 93.25±0.20 | 92.00±0.19 | 94.21±0.32 |
| Reuters | 46.27±0.05 | 40.02±2.40 | 50.10± 0.00 | 44.10±0.01 | 43.70±1.13 | 54.04±0.56 | 59.46±1.00 | 42.17±0.35 | 60.55±2.10 |
| Football | 82.48±1.01 | 78.98±1.29 | 77.08± 2.55 | 77.42±0.00 | 75.40±2.26 | 91.57±0.93 | 89.65±0.10 | 85.21±0.54 | 91.20±1.90 |
| ANIMAL | 58.17±2.00 | 45.92±1.80 | 57.96±1.33 | 50.15±0.00 | 32.61±1.81 | 66.16±0.54 | 71.84±1.20 | 59.21±0.68 | 73.41±1.00 |
| NoisyMNIST | 56.90±0.05 | 53.42±0.31 | 64.27±0.50 | 60.43±0.25 | 43.21±1.20 | 80.50±0.31 | 81.47±0.05 | 73.18±0.50 | 82.50±2.50 |

### 3.1 EXPERIMENTAL RESULTS AND ANALYSIS

In this section, we show the clustering accuracy of EKNN and comparison approaches on different datasets based on four metrics. Note that we list the clustering results of EKNN and its extension (EKNN for multi-view subspace learning) on different datasets. For comparison, we show the clustering performance of SSSL-M without supervisory information. According to Tables I-IV, we can draw some interesting insights as follows:

Table 2: Clustering performance (NMI%±STD%) on different datasets.

| Data sets | COMIC | DMF | MVEC | BMVC | DiMSC | RMSL | SSSL-M | Ours | Ours(extension) |
|---|---|---|---|---|---|---|---|---|---|
| COIL20 | 78.47±1.00 | 85.80±0.26 | 87.76±0.89 | 80.07±0.00 | 84.61±1.75 | 94.10±1.32 | 87.43±0.05 | 82.45±0.50 | 89.20±0.42 |
| ORL | 91.20±0.05 | 86.27±1.00 | 80.23±1.16 | 85.20±0.00 | 94.02±1.35 | 94.96±0.47 | 91.47±1.00 | 88.39±0.25 | 95.10±0.50 |
| BBCSprot | 90.23±0.21 | 53.39±0.08 | 90.32±0.02 | 70.80±0.00 | 85.11±0.13 | 91.73±0.52 | 94.51±0.50 | 86.50±0.80 | 94.70±1.00 |
| Reuters | 36.50±2.31 | 22.88±1.15 | 30.17±0.02 | 25.41±0.00 | 23.31±0.33 | 37.49±0.46 | 41.55±0.05 | 31.50±0.47 | 42.90±1.28 |
| Football | 89.57±0.01 | 83.38±0.79 | 83.36±1.11 | 80.22±0.00 | 82.16±1.45 | 92.29±0.42 | 89.78±0.05 | 80.15±2.00 | 90.25±0.79 |
| ANIMAL | 73.02±0.22 | 56.64±1.31 | 68.72±0.50 | 66.76±0.00 | 44.62±0.89 | 73.19±0.60 | 77.49±1.00 | 70.76±0.90 | 79.00±1.92 |
| NoisyMNIST | 79.42±0.01 | 62.48±0.35 | 72.41±0.31 | 71.23±1.20 | 57.36±0.05 | 84.28±0.59 | 85.91±1.21 | 72.30±0.48 | 86.20±0.95 |

Table 3: Clustering performance (F-score%±STD%) on different datasets.

| Data sets | COMIC | DMF | MVEC | BMVC | DiMSC | RMSL | SSSL-M | Ours | Ours(extension) |
|---|---|---|---|---|---|---|---|---|---|
| COIL20 | 70.05±0.05 | 63.22±1.03 | 64.05±2.12 | 67.62±0.00 | 71.99±0.50 | 81.20±1.72 | 75.89±0.05 | 76.39±1.50 | 77.10±0.50 |
| ORL | 75.98±2.00 | 64.61±2.56 | 61.94±2.31 | 62.95±0.00 | 80.71±1.38 | 84.22±1.43 | 83.26±0.59 | 82.60±0.39 | 84.20±1.40 |
| BBCSprot | 85.23±0.01 | 62.46±0.01 | 93.72±0.02 | 80.81±0.00 | 91.02±0.14 | 95.35±0.41 | 91.45±0.05 | 90.48±0.27 | 92.15±0.95 |
| Reuters | 40.20±1.01 | 34.63±1.37 | 39.59±0.02 | 37.93±0.00 | 33.01±0.39 | 44.20±0.45 | 46.84±0.01 | 40.29±0.39 | 48.21±2.34 |
| Football | 71.90±2.00 | 70.35±0.87 | 67.08±3.70 | 63.72±0.00 | 67.13±1.19 | 83.87±1.57 | 79.14±1.45 | 75.28±1.14 | 79.47±0.62 |
| ANIMAL | 54.80±1.00 | 32.86±2.17 | 49.31±2.49 | 41.47±0.00 | 20.66±1.10 | 57.29±1.14 | 58.76±0.05 | 42.16±0.33 | 60.27±2.84 |
| NoisyMNIST | 53.42±0.01 | 33.42±0.10 | 50.41±0.04 | 47.31±0.01 | 32.40±0.10 | 70.05±0.15 | 70.16±0.01 | 52.50±0.27 | 71.30±1.18 |

- The proposed EKNN and its extension can achieve satisfied clustering performance on multi-view datasets. The clustering results of the proposed EKNN are not as well as its extension (EKNN for multi-view subspace learning). It can be explained by the fact that more complex mappings are able to result in more desired result, i.e., introducing the latent representation and shared subspace representation in the final objective function.

- On different multi-view datasets, the proposed EKNN still obtain relatively better results than some multi-view clustering methods, which can be explained by the effectiveness of dividing the multi-view clustering into three sub-problems based on $K$-means or kernel $K$-means. Besides, utilizing linear clustering as one part in the three-level optimization problem is able to reduce the overfitting of the final clustering to some degree and indirectly increase the clustering performance.

- The proposed EKNN and its extension consistently achieve satisfied clustering performances on all datasets in terms of different metrics, which demonstrate their robustness to multiple types of data and the effectiveness of dividing the multi-view clustering into three sub-problems based on clustering or learning, i.e., 1) linear clustering or learning on the set of data points in the dataset, 2) nonlinear clustering or learning on the set of linear clusters obtained by 1), and 3) multi-view clustering or learning by incorporating partition matrices from different views obtained by linear and nonlinear clustering or learning.

## 3.2 Influence for number of liner clusters

In order to evaluate the impact for the number of linear clusters $p$ on different datasets, we vary it in the range of $\{20, 40, 60, 80, 100, 120, 140, 160, 180, 200\}$ for the proposed EKNN. According to Fig. 3, we can observe that the clustering results in terms of different metrics increase with the increasing value of $p$ and values of these four metrics increase slowly when they reach a certain value. However, the computing cost also increases with the increasing value of $p$ based on the complexity analysis of our algorithm. We then choose $p = 160$ for the proposed EKNN in the experiment.

## 3.3 Running Time

We also present the running time of the proposed EKNN and its extension on Table V, which is conducted on different datasets. Some conclusions can be obtained as follows:

- The proposed EKNN and its extension are able to own satisfied time costs on with different number of liner clusters, which validates their efficiencies in the experiment on different datasets. Due to the efficiency, EKNN uses much less time cost compared with RMSL and SSSL-M.

Table 4: Clustering performance (RI%±STD%) on different datasets.

| Data sets | COMIC | DMF | MVEC | BMVC | DiMSC | RMSL | SSSL-M | Ours | Ours(extension) |
|---|---|---|---|---|---|---|---|---|---|
| COIL20 | 96.20±0.21 | 95.44±0.19 | 95.35±0.49 | 96.66±0.00 | 97.14±0.11 | 97.90±0.41 | 97.18±0.15 | 97.28±0.50 | 97.92±1.20 |
| ORL | 97.84±0.01 | 98.31±0.14 | 97.16±0.65 | 98.23±0.00 | 98.97±0.05 | 99.15±0.07 | 98.42±0.20 | 98.30±0.37 | 98.70±0.90 |
| BBCSprot | 95.14±0.01 | 82.45±0.00 | 97.02±0.01 | 90.23±0.00 | 95.72±0.10 | 97.81±0.19 | 94.56±1.52 | 95.10±0.59 | 97.90±1.20 |
| Reuters | 68.99±0.17 | 58.71±1.29 | 74.17±0.62 | 69.15±0.00 | 67.49±0.28 | 71.37±0.35 | 73.81±0.17 | 69.48±1.35 | 74.12±0.85 |
| Football | 97.00±0.01 | 96.51±0.44 | 96.22±0.54 | 96.69±0.00 | 96.74±0.59 | 98.40±0.16 | 97.68±0.50 | 96.25±0.47 | 97.95±0.72 |
| ANIMAL | 97.25±0.35 | 97.09±0.22 | 97.12±0.25 | 96.98±0.00 | 96.30±0.23 | 98.12±0.05 | 98.87±1.00 | 97.10±0.91 | 98.95±1.82 |
| NoisyMNIST | 84.00±0.27 | 84.05±0.01 | 84.20±0.05 | 83.76±0.10 | 83.52±1.05 | 83.16±0.35 | 84.20±0.15 | 83.45±1.20 | 85.10±0.17 |

Table 5: Clustering speed (s) of EKNN and its extension on all dataset with different $p$.

| Method | COIL20 | ORL | BBCSport | Reuters | Football | ANIMAL | NoisyMNIST |
|---|---|---|---|---|---|---|---|
| RMSL | 5210.0 | 3200.0 | 3670.5 | 3001.5 | 4200.0 | 1640.6 | 32000.0 |
| SSSL-M | 5560.6 | 3590.0 | 3700.5 | 3130.5 | 4250.0 | 1720.5 | 32406.7 |
| Ours[p=120] | 435.2 | 256.3 | 359.7 | 292.5 | 407.1 | 1432.2 | 3125.4 |
| Ours(extension)[p=120] | 672.4 | 432.8 | 589.2 | 789.0 | 639.0 | 2345.1 | 7639.0 |
| Ours[p=140] | 528.9 | 423.2 | 638.2 | 390.5 | 542.1 | 1934.2 | 4210.5 |
| Ours(extension)[p=140] | 830.4 | 672.9 | 723.8 | 920.5 | 798.2 | 3410.5 | 8219.8 |
| Ours[p=160] | 634.6 | 590.3 | 792.8 | 414.9 | 720.5 | 2319.5 | 4720.7 |
| Ours(extension)[p=160] | 926.6 | 825.9 | 904.2 | 690.3 | 1025.2 | 4021.6 | 8935.0 |
| Ours[p=180] | 890.4 | 746.2 | 880.5 | 530.2 | 843.1 | 2789.0 | 5290.5 |
| Ours(extension)[p=180] | 1204.3 | 1190.4 | 1068.0 | 725.6 | 1205.6 | 4529.0 | 9534.1 |

- The proposed EKNN needs less time costs than its extension on all datasets. It can be explained by the fact that its extension introduces more regularization terms for the effectiveness. Therefore, we can observe that the proposed EKNN and its extension can well balance the efficiency and effectiveness of multi-view clustering on different datasets.

## 3.4 CONVERGENCE ANALYSIS

We also show the convergence analysis of the proposed EKNN on COIL20, ORL and BBCSport dataset in Fig. 4. It can be observed that EKNN can achieve convergence in some iterations, which further demonstrates the feasibility of EKNN.

## 4 CONCLUSIONS

In this work, we have shown that multi-view clustering can be regarded as a three-level optimization problem. We flexibly divide the multi-view clustering into three sub-problems based on $K$-means or kernel $K$-means, i.e., the linear clustering on the set of original data points, the nonlinear clustering of each view on the set of linear clusters, and multi-view clustering by incorporating partition matrices from different views obtained by linear and nonlinear clustering. The proposed EKNN unifies three different sub-problems into a framework and we use the iterative algorithm to optimize the formulated problem. EKNN is able to effectively and rapidly cluster data points from different views. We also extend EKNN for multi-view clustering to the case of multi-view subspace learning, which is able to learn a satisfied latent representation shared by different views. Experiments on different datasets also demonstrate the effectiveness and efficiency of EKNN in terms of different metrics.

### ACKNOWLEDGMENTS

This work was supported by Eastern Talent Plan Leading Project under Grant BJKJ2024011 and National Natural Science Foundation of China (62402303).

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

# A APPENDIX

## A.1 RELATED WORK OF $K$-MEANS ALGORITHM AND ANALYSIS

As one of the most efficient algorithms for clustering, $K$-means algorithm uses an initial set of cluster centers as the start and decreases the sum of squared errors by iteratively refining this set, which has gained considerable attentions in the related literature. The $K$-means algorithm aims to minimize the objective function $P$ as:

$$P(S, V) = \sum_{l=1}^{k} \sum_{x_i \in S_l} d(x_i, v_l),\tag{11}$$

where $S = \{S_1, ... S_k\}$ denotes a partition of the dataset $X$, $k$ is the number of real clusters in $X$, $S_l \subset X, \cup_{l=1}^{k} S_l = X, S_l \cap S_j = \varnothing$ for $1 \leq l \neq j \leq k$, $V = \{v_l\}_{l=1}^{k}$ and $v_l$ indicates the $l$-th center of $S_l$. According to Eq. (11), we can observe that $K$-means algorithm is related to two updated equations based on $V$ and $S$. The alternative optimization method is used to seek for the optimal minimized value of $P$ and the updated equations are shown in the following. Given $V$, we update $S$ by

$$x_i \in S_l, \ \ if \ \ l^* = \arg\min_l d(x_i, v_l),\tag{12}$$

where $1 \leq i \leq n$, $n$ is the number of data points and $1 \leq l \leq k$. Given $S$, we update $V$ by

$$v_l = \frac{\sum_{x_i \in S_l} x_i}{|S_l|},\tag{13}$$

where $1 \leq l \leq k$. The time complexity is $O(nkt)$ for $K$-means algorithm with $t$ being the number of total iterations.

Based on Eq. (11), we can find that the cost of $K$-means algorithm is much lower than that of calculating the distances between all data points since it just needs to calculate the distances between centers and data points. We also observe that the clustering performance of $K$-means is influenced by some factors, i.e., the initial cluster centers. Different initial cluster centers usually result in different results. However, $K$-means algorithm often fails to achieve satisfied clustering performance when clusters are nonlinearly separable.

## A.2 MULTI-VIEW REPRESENTATION LEARNING

Multi-view representation learning aims to effectively explore the complete and consistent information from different views. Various multi-view learning methods Kumar et al. (2011); Kumar & III (2011); Xu et al. (2015); Zhang et al. (2017); Yang et al. (2019) have been proposed by finding the consistent hypotheses for clustering across different views. For example, canonical correlation analysis (CCA) Hotelling (1935), kernelized CCA Akaho (2006), deep neural networks based CCA Andrew et al. (2013), and deep canonically correlated autoencoder (DCCAE) Wang et al. (2015) are representative methods in multi-view representation learning, which have achieved great success. To be specific, CCA minimizes the correlation between two different views to find a shared representation. CCA is formulated as:

$$(P^1, P^2) = \arg\min_{P^1, P^2} tr((P^1)^T X^1 (X^2)^T P^2),$$
$$s.t. \ (P^v)^T X^v (X^v)^T P^v = I, \ v = 1, 2,\tag{14}$$

where $X^v = [x_1^v, x_2^v, ..., x_n^v] \in R^{m \times n}$ denotes the representation of the $v$-th view. $n$ and $m$ represent the size of dataset and the dimension of the $v$-th view, respectively. $k$ indicates the dimension for the shared representation. $P^v$ and $I$ denote the mapping matrix and the identity matrix of the $v$-th view, respectively. As one method based on CCA, Zhang *et al.* Cao et al. (2015) presented to exploit the correlations among multiple views as:

$$(P^1, ..., P^V) = \arg\min_{P^1, ..., P^V} \sum_{v=1}^{V} tr(P^v X^v L^v (X^v)^T (P^v)^T)$$
$$+ \lambda \sum_{\neq u} HSIC(P^v X^v, P^u X^u),\tag{15}$$
$$s.t. \ (P^v)^T X^v (X^{(v)})^T P^v = I, \ v = 1, 2, ..., V,$$

where $V$ denotes the number of total views and $L^v$ is the graph Laplacian matrix for the $v$-th view, respectively. Li *et al.* Li et al. (2019) studied complex relations among multiple views and learned the underlying latent representation. It is formulated by:

$$
\begin{aligned}
\min_{\Theta, H} \alpha L_E(\{\Theta_S^v\}_{v=1}^V, H; \{\Theta_E^v\}_{v=1}^V) + \gamma R(\{\Theta_E^v\}_{v=1}^V) \\
+ L_S(\{X^v\}_{v=1}^V, H; \{\Theta_S^v\}_{v=1}^V, \Theta_C) + \beta R(\{\Theta_S^v\}_{v=1}^V, \Theta_C),
\end{aligned}
\tag{16}
$$

where $L_E(.;.)$ is the loss of reconstruction to update $H$, $R(.)$ is adopted for representing the term for regularization, $L_S(.;.)$ denotes the loss based on self-representation, $\Theta$ including $\Theta_{E^v}$, $\Theta_{S^v}$ and $\Theta_C$ are parameters of the model.

Different from multi-view representation learning, cross-view representation learning targets at searching for mappings between two different views. Numerous methods based on cross-view representation learning have been proposed Rasiwasia et al. (2010); Chung et al. (2018); Castrejón et al. (2016); Zhou et al. (2019). The embedding spaces of two different views are simultaneously learned and aligned by adversarial training Chung et al. (2018). To solve partially view-aligned problem, the literature in Yang et al. (2021) simultaneously learns the representation and aligns the data by a contrastive loss which is robust to noise. The major difference between the above methods and our work is that they do not take the efficiency for multi-view clustering into consideration while our method is able to achieve final clustering results with desired efficiency.

### A.3 SOLUTION FOR THE FORMULATED PROBLEM

To make the objective function of EKNN in Eq. (6) separable, we introduce the auxiliary variable $R^v$ to replace $\Theta^v$ and obtain the equivalent objective function as follows:

$$
\begin{aligned}
\min_H F = & \sum_{v=1}^V \alpha ||\Theta^v - W^v V^v||_F^2 + \beta Tr(K^v) \\
& - \beta Tr((\hat{U}^v)^T K^v \hat{U}^v) + \gamma ||W^v \hat{U}^v - HG^v||_F^2 \\
& + \eta ||X^v - X^v \Theta^v||_F^2 + \mu ||R^v||_*, \\
s.t. \ & w_{ij}^v \in \{0,1\}, \ \sum_{j=1}^p w_{ij}^v = 1, \ 1 < \sum_{i=1}^n w_{ij}^v < n, \\
& u_{ij}^v \in \{0,1\}, \ \sum_{j=1}^k u_{ij}^v = 1, \ 1 < \sum_{i=1}^p u_{ij}^v < p, \\
& h_{ij}^v \in \{0,1\}, \ \sum_{j=1}^k h_{ij}^v = 1, \ 1 < \sum_{i=1}^n h_{ij}^v < n, \\
& \Theta^v = R^v.
\end{aligned}
\tag{17}
$$

We then divide the problem in Eq. (17) into different sub-problems to minimize:

- $\min_{W^v, V^v} F$ with fixed $U^v$, $R^v$, $\Theta^v$, $H$ and $G^v$.
- $\min_{U^v} F$ with fixed $W^v$, $R^v$, $V^v$, $H$, $\Theta^v$ and $G^v$.
- $\min_{H, G^v} F$ with fixed $W^v$, $R^v$, $\Theta^v$, $V^v$ and $U^v$.
- $\min_{\Theta^v} F$ with fixed $U^v$, $R^v$, $W^v$, $V^v$, $H$ and $G^v$.
- $\min_{R^v} F$ with fixed $U^v$, $\Theta^v$, $W^v$, $V^v$, $H$ and $G^v$.

These optimization sub-problems are solved in the following.

**(1) $W^v, V^v$-sub-problem:**

Given $U^v$, $R^v$, $\Theta^v$, $H$ and $G^v$, $\min_{W^v, V^v} F$ is formulated as:

$$
\min_{W^v, V^v} \alpha ||\Theta^v - W^v V^v||_F^2 + \gamma ||W^v \hat{U}^v - HG^v||_F^2.
\tag{18}
$$

We then simplify Eq. (18) as:

$$\min_{W^v, \tilde{V}^v} ||\tilde{\Theta}^v - W^v \tilde{V}^v||_F^2, \tag{19}$$

where $\tilde{\Theta}^v = [\alpha^{\frac{1}{2}}\Theta^v, \gamma^{\frac{1}{2}}HG^v]$ is obtained by concatenating $\Theta^v$ and $HG^v$, and $\tilde{V}^v = [\alpha^{\frac{1}{2}}V^v, \gamma^{\frac{1}{2}}\hat{U}^v]$. Therefore, we can regard $\tilde{\Theta}^v$ and $\tilde{V}^v$ as the new representation of $\Theta^v$ and the corresponding cluster center matrix, respectively. Then the optimization problem in Eq. (18) is transformed into a $K$-means clustering problem regarding $\tilde{\Theta}^v$ and we solve it by the update rules of $W^v$ and $\tilde{V}^v$. The details are shown in the following.

Given $\tilde{V}^v$, we update $W^v$ by

$$w_{ij}^v = \begin{cases} 1, & \text{for } j = \arg\min ||\tilde{\theta}_i^v - \tilde{v}_l^v||^2, & (20) \\ 0, & \text{otherwise}, & (21) \end{cases}$$

where $l \in [1, p]$, $\tilde{v}_l^v$ is the $l$-th row of $\tilde{V}^v$ and $\tilde{\theta}_i^v$ denotes the $i$-th row of $\tilde{\Theta}^v$. Given $W^v$, we update $\tilde{V}^v$ as

$$\tilde{v}_j^v = \frac{\sum_{i=1}^n w_{ij}^v \tilde{\theta}_i^v}{\sum_{i=1}^n w_{ij}^v}. \tag{22}$$

Then the update rules of $W^v$ and $\tilde{V}^v$ can be obtained in this manner.

**(2) $U^v$-sub-problem:**

Given $\Theta^v$, $R^v$, $W^v$, $V^v$, $H$ and $G^v$, $\min_{U^v} F$ is formulated by

$$\min_{U^v} \beta(Tr(K^v) - Tr(\hat{U}^v K^v \hat{U}^v)) + \gamma ||W^v \hat{U}^v - HG^v||_F^2. \tag{23}$$

We further rewrite Eq. (23) by

$$\min_{U^v} \beta Tr(K^v) - Tr((\hat{U}^v)^T L^v \hat{U}^v), \tag{24}$$

where $L^v = \beta K^v + \gamma(W^v)^T(I_n - \hat{H}^v(\hat{H}^v)^T)W^v$ and $I_n \in R^{n \times n}$ is the identity matrix. It has been proved that the equivalence between kernel $K$-means and spectral clustering algorithm Dhillon et al. (2007). Thus, Eq. (24) is equivalent to the problem as follows:

$$\max_{\hat{U}^v} Tr((\hat{U}^v)^T L^v \hat{U}^v), \ s.t. \ (\hat{U}^v)^T \hat{U}^v = I_k, \tag{25}$$

Therefore, the spectral clustering method can be used to solve this problem.

**(3) $H, G^v$-sub-problem:**

Given $\Theta^v$, $R^v$, $W^v$, $V^v$ and $U^v$, $\min_{H,G^v} F$ is formulated as:

$$\min_{H,G^v} \sum_{v=1}^V ||W^v \hat{U}^v - HG^v||_F^2. \tag{26}$$

Likewise, we can solve this problem as the paradigm of $K$-means. Given $\Theta^v$, $W^v$, $R^v$, $V^v$, $H$ and $U^v$, we update $G^v$ as

$$g_l^v = \frac{\sum_{i=1}^n h_{il} q_i^v}{\sum_{i=1}^n h_{il}}, \tag{27}$$

where $q_i^v$ denotes the $i$-th row of $W^v \hat{U}^v$. Given $\Theta^v$, $R^v$, $W^v$, $V^v$, $G^v$ and $U^v$, we update $H$ as

$$h_{il} = \begin{cases} 1, & \text{for } l = \arg\min \sum_{v=1}^V ||q_i^v - g_e^v||^2 & (28) \\ 0, & \text{otherwise}, & (29) \end{cases}$$

where $e \in [1, k]$, $l \in [1, k]$ and $i \in [1, n]$.

**(4) $\Theta^v$-sub-problem:**

Given $U^v$, $W^v$, $R^v$, $V^v$, $H$ and $U^v$, $\min_{H, G^v} F$ is formulated by:

$$\min_{\Theta^v} F = \sum_{v=1}^{V} \alpha ||\Theta^v - W^v V^v||_F^2 + \eta ||X^v - X^v \Theta^v||_F^2,$$

$$s.t. \quad \Theta^v = R^v. \tag{30}$$

We then adopt the Augmented Lagrange Multiplier (ALM) Lin et al. (2011) to tackle this problem as follows:

$$\min_{\Theta^v} F = \sum_{v=1}^{V} \alpha ||\Theta^v - W^v V^v||_F^2 + \eta ||X^v - X^v \Theta^v||_F^2$$

$$+ \Phi(Y^v, \Theta^v - R^v), \tag{31}$$

where $\Phi(Y^v, \Theta^v - R^v) = \frac{\rho}{2} ||\Theta^v - R^v||_F^2 + \langle Y^v, \Theta^v - R^v \rangle$, $\langle ., . \rangle$ denotes the Frobenius inner product, $\rho > 0$ and $Y^v$ is the Lagrange multiplier. We take the derivative with respect to $\Theta^v$ and then set it to be zero. Then the closed-form solution can be obtained as:

$$\Theta^v = \left[ 2\eta (X^v)^T X^v + (2\alpha + \rho) I \right]^{-1}$$

$$\cdot \left[ 2\eta (X^v)^T X^v + \rho R^v - Y^v + 2\alpha W^v V^v \right], \tag{32}$$

where $I$ denotes the identity matrix.

**(5) $R^v$-sub-problem:**

Given $\Theta^v$, $W^v$, $H$, $V^v$, $H$ and $U^v$, $\min_{R^v} F$ is formulated by:

$$\min_{\Theta^v} F = \sum_{v=1}^{V} \mu ||R^v||_*, \quad s.t. \quad \Theta^v = R^v. \tag{33}$$

Then we can obtain

$$R^v = \arg \min_{R^v} \frac{\mu}{\rho} ||R^v||_* + \frac{1}{2} ||R^v - (\Theta^v + \frac{Y^v}{\rho})||_F^2. \tag{34}$$

The Lagrange multiplier $Y^v$ is updated by:

$$Y^v = Y^v + \rho(\Theta^v - R^v). \tag{35}$$

Since subproblems for $\Theta^v$, $R^v$ and $Y^v$ are adopted to obtain effective subspace representation of $X^v$, we can regard them as the same subproblem for learning $W^v$ and $V^v$ based on linear clustering. Then the objective function $F$ of EKNN can be approximately minimized by iteratively solving these sub-problems. The whole algorithm is presented in **Algorithm 1**.

---

**Algorithm 1:** Algorithm of the proposed method

---

**Input:** Multi-view dataset $X$, number of real clusters $k$ and max iteration $s$
**Output:** Clustering result $H$
**Initialize:** Randomly initialize $\Theta, W, U, V, R, Y, H, \rho = 10^{-6}$ and $G$, $i = 0$
**repeat**
    Update $W, V$ with the fixed $\Theta, R, U, H$ and $G$;
    Update $U$ with the fixed $\Theta, R, W, V, H$ and $G$;
    Update $H, G$ with the fixed $\Theta, W, R, V$ and $U$;
    Update $\Theta$ with the fixed $U, W, R, V, H$ and $G$;
    Update $R, Y$ with the fixed $U, W, V, \Theta$ $H$ and $G$;
    $i = i + 1$;
**until** $i > s$;

---

It can be observed that the efficiency, effectiveness, completeness and consistency of EKNN are determined by $\alpha$, $\beta$ and $\gamma$. The following two cases can be obtained according to the choices of these three parameters. (1) We can regard the clustering result as linear multi-view clustering based on $K$-means if $V > 1$, $\alpha \neq 0$, $\beta = 0$ and $\gamma \neq 0$. (2) The clustering result can be seen as multi-view clustering based on kernel $K$-means if $V > 1$, $\beta \neq 0$ and $\gamma \neq 0$.

## A.4 Complexity Analysis

The time complexity of EKNN consists of different parts according to the objective function in Eq. (6). To be specific, the complexity of solving $W^v, V^v$-sub-problem is $O(npsV(m+k))$, where $s$ is the number of total iterations for this clustering. It needs $O((k+m)p^2V)$ to find the solution of $U^v$-sub-problem. The complexity to solve $H, G^v$-sub-problem is $O(nk^2sV)$. Therefore, the total time complexity of solving these three subproblems is $O(npsV(m+k)+(k+m)p^2V+nk^2sV)$. The complexities to update $\Theta^v$, $Y^v$ and $R^v$ are $O(n^3)$. In general, the total computational complexity including employing subspace representation $\Theta^v$ is $O(n^3)$.

## A.5 Datasets

We adopt seven different datasets in the experiment for comprehensive evaluation, i.e., ORL face images Samaria & Harter (1994), COIL20 object images Nene et al. (1996), Reuters multilingual dataset Amini et al. (2009), Football Li et al. (2019), BBCSport documentsGreene & Cunningham (2006), ANIMAL Lampert et al. (2014) and NoisyMNIST Lin et al. (2021).

- **ORL** consists of 40 diverse subjects and each subject has ten face images. It has total three views, which are obtained by different types of features.
- **COIL20** contains over 20 objects and it has 1440 images obtained by the camera with multiple angles. This dataset is characterized by three views.
- **Reuters** has six classes and it is a textual dataset with 1000 different newswire, which are expressed in five languages, i.e., French, Spanish, German, Italian and English.
- **Football** has 20 clubs from English Premier League in BBC website and there are total 248 football players in the league. This dataset has total nine different views.
- **BBCSport** is taken from sports articles and it has 544 documents, which consists of two different views in five topical areas.
- **ANIMAL** consists of 50 different animal classes and it has total 30475 images. We use VGG19 Simonyan & Zisserman (2014) and DECAF Krizhevsky et al. (2012) to extract two types of deep features, which are used as two different views.
- **NoisyMNIST** has two views with selected MNIST images containing Gaussian noise in one view and 70k images from MNIST dataset in the second view. We use 10k validation images and 10k testing image in NoisyMNIST for the experiment as in Lin et al. (2021).

## A.6 Compared methods

We conduct experiments to demonstrate the effectiveness of EKNN for multi-view clustering by comparing ours with some existing representative methods in the following.

- **DiMSC** Cao et al. (2015) incorporates complementary information from multiple views by making use of the diversity among different subspace representations.
- **LT-MSC** Zhang et al. (2015) employs the low-rank tensor for exploring high-order relationships among different views.
- **DMF** Zhao et al. (2017) enforces the nonnegative representation of each view in the last layer to be consistent by maximizing the mutual information from different views.
- **MVEC** Tao et al. (2017) produces the basic clustering result for each view and then learns a shared one based on the obtained basic clustering results.
- **RSML** Li et al. (2019) comprehensively describes the data and flexibly encodes complementary information from multiple views. It utilizes backward encoding networks to encode specific subspace representations from different views into the latent representation.
- **COMIC** Peng et al. (2019) maps the data into a space, where cluster assignment consistency and geometric consistency are simultaneously considered.
- **SSSL-M** Qin et al. (2022a) builds an indicator matrix which is anti-block-diagonal for pursuing the block-diagonal structure of the shared affinity matrix with small amount of supervisory information.

We adopt four commonly used metrics for comprehensive investigations, i.e., Accuracy (ACC), F-score, Normalized Mutual Information (NMI), and Rand Index (RI), which are adopted to reflect the clustering performance of the proposed method. The higher values of these metrics indicate more satisfied clustering performance. We conduct experiments on AMD Ryzen5 2600 with 16G RAM. To reduce the randomness, we run each experiment for 30 times and list the mean as well as the standard deviation (STD) as the final result. We adopt Gaussian kernel in the experiment for comparison and analysis.

### A.7  PARAMETER SELECTION

The proposed EKNN consists of some parameters, i.e., balance weights $\alpha$, $\beta$, $\gamma$, $\eta$ and $\mu$. The grid search strategy is employed to find the optimal parameters in this work. To study the influences of balance weights $\alpha$, $\beta$, $\gamma$, $\eta$ and $\mu$, we change their values in $\{0.2, 0.4, 0.6, 0.8, 1\}$. Based on Figs. 5-9, it is observed that our method is relatively robust to the choices of $\alpha$, $\beta$, $\gamma$, $\eta$ and $\mu$ when their values are in $\{0.6, 0.8, 1\}$. Moreover, we can remarkably improve the final clustering performance of the proposed method with good choices of $\alpha$, $\beta$, $\gamma$, $\eta$ and $\mu$. The optimal clustering performance can be obtained when $\alpha = 0.8$, $\beta = 0.8$, $\gamma = 0.8$, $\eta = 0.8$ and $\mu = 0.6$. Likewise, the values of parameters in the extension of EKNN can also be obtained in the same manner and we omit here for simplicity. For parameters in the extension version, we set $\alpha = 0.8$, $\beta = 0.8$, $\gamma = 0.8$, $\eta = 0.8$, $\eta^{'} = 0.8$, $\mu = 0.6$ and $\mu = 0.6$.

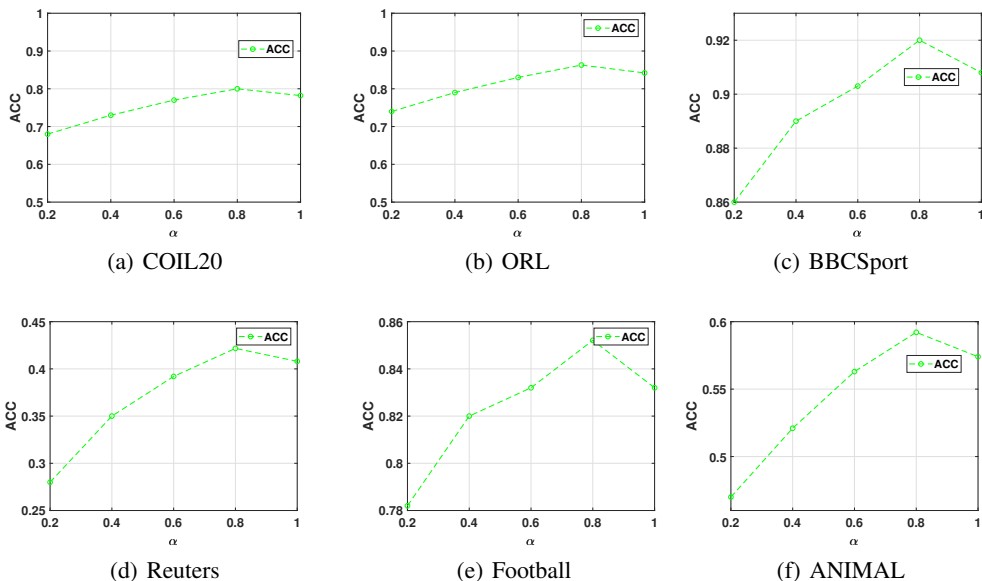

Figure 5: Clustering performance of EKNN with different $\alpha$ on different datasets.

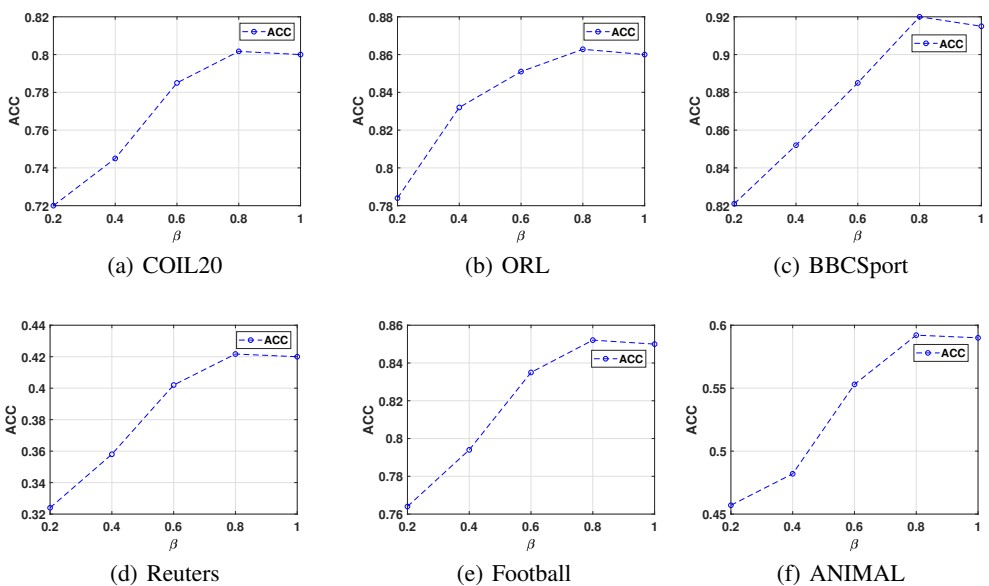

Figure 6: Clustering performance of EKNN with different $\beta$ on different datasets.

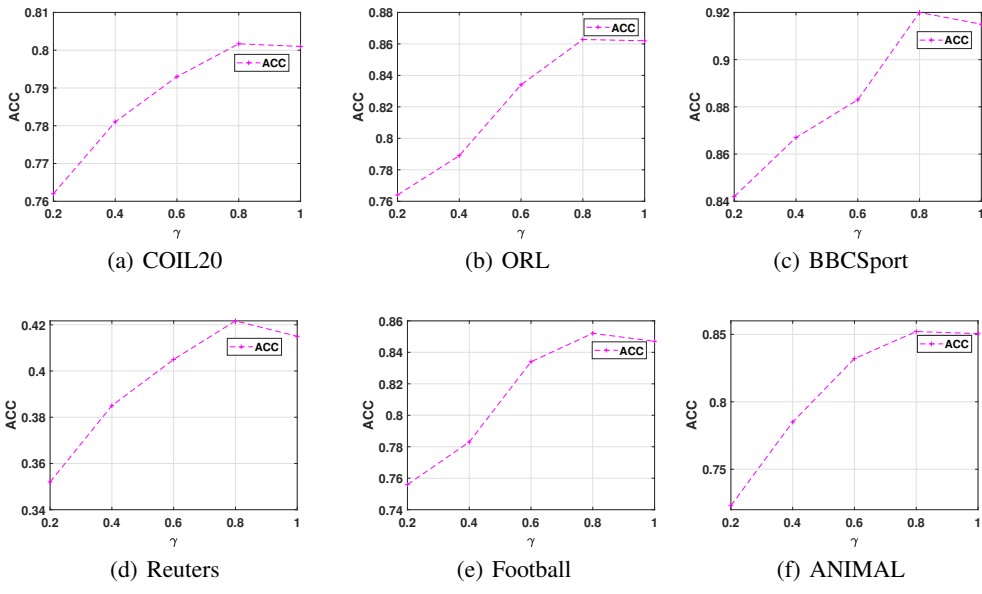

Figure 7: Clustering performance of EKNN with different $\gamma$ on different datasets.

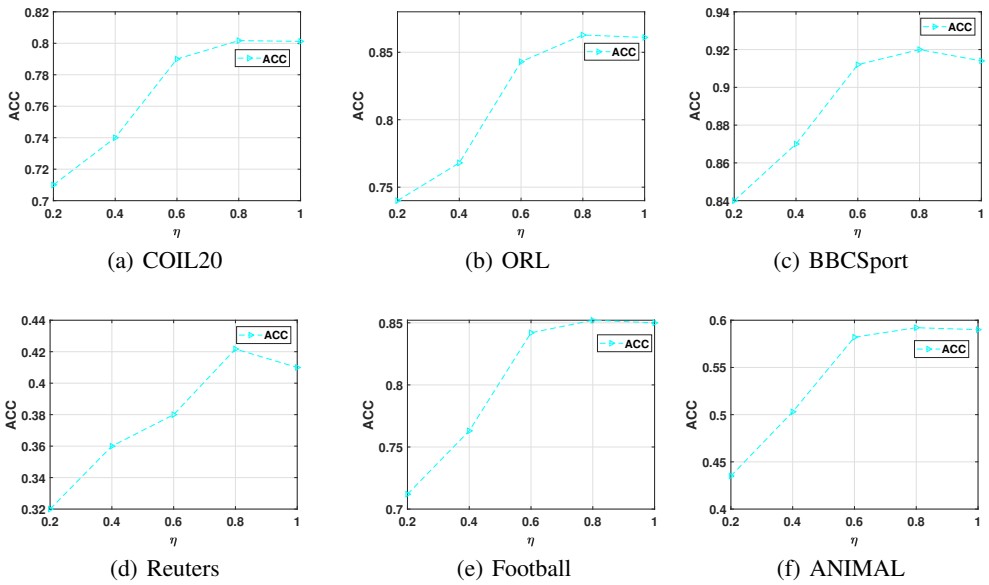

Figure 8: Clustering performance of EKNN with different $\eta$ on different datasets..

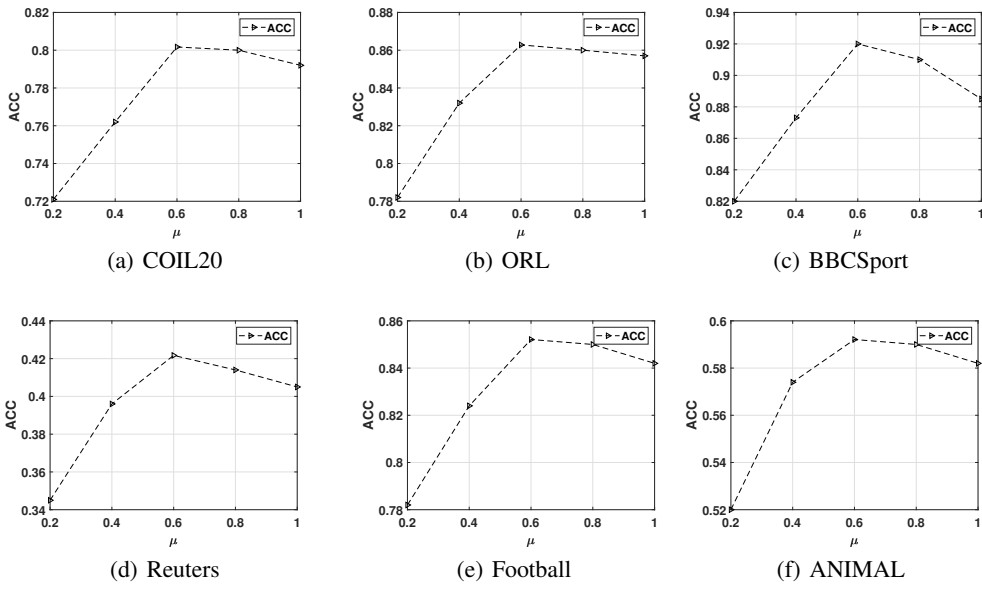

Figure 9: Clustering performance of EKNN with different $\mu$ on different datasets.

