# OpenReview forum: "Explainable $ K $-means Neural Networks for Multi-view Clustering"
_ICLR.cc/2026/Conference — ICLR 2026 Poster_

### Official Review · Reviewer_xM3h · 2025-10-23

**Soundness:** 3
**Presentation:** 3
**Contribution:** 3
**Rating:** 8
**Confidence:** 5

**Summary:**

This work gives a novel insight to the multi-view clustering community, i.e., the multi-view clustering can be regarded as a three-level optimization problem. The authors propose to divide the multi-view clustering into three sub-problems: the linear clustering on the set of original data points based on $ K $-means, the nonlinear clustering of each view on the set of linear clusters based on kernel $ K $-means, and multi-view clustering by integrating information from different views for the set of partitions in terms of $ K $-means.

**Strengths:**

The authors propose to divide the multi-view clustering into three sub-problems: the linear clustering on the set of original data points based on $ K $-means, the nonlinear clustering of each view on the set of linear clusters based on kernel $ K $-means, and multi-view clustering by integrating information from different views for the set of partitions in terms of $ K $-means.

**Weaknesses:**

1. Section 3 describes the proposed method. It is not easy to follow, partly because of the necessary notations for multi-view data, but also because the motivation of the approaches is not well explained. For example, why one combines Wv and Uˆv together in Eq. (10), where Wv comes from the linear clustering (7), while Uˆv comes from the nonlinear clustering (9).
2. Following the above point, the motivation of the proposed approach also needs further justification. For example, Eq. (8) applies K-means on the subspace representation. Why one needs to apply K-means on the subspace representation instead of directly applying subspace clustering techniques?
3. Also, Eq. (9) then applies kernel K-means on the K-means coefficients of the above approach. What is the motivation behind this step?
4. Why do we need to further cluster the coefficients?

**Questions:**

Following the above point, the motivation of the proposed approach also needs further justification. For example, Eq. (8) applies K-means on the subspace representation. Why one needs to apply K-means on the subspace representation instead of directly applying subspace clustering techniques?

---

> ### Author Response · Authors · 2025-11-17
>
> Q1: Section 3 describes the proposed method. It is not easy to follow, partly because of the necessary notations for multi-view data, but also because the motivation of the approaches is not well explained. For example, why one combines Wv and Uˆv together in Eq. (10), where Wv comes from the linear clustering (7), while Uˆv comes from the nonlinear clustering (9).
>
> A1: Thanks for the comment. As reviewer pointed, we have improved presentation in the motivation and necessary notations for multi-view data. The motivation of this paper is that most existing work fails to simultaneously balance the effectiveness, efficiency, consistency and completeness of nonlinear multi-view clustering. We aim to solve this issue and propose to divide the nonlinear multi-view clustering into three sub-problems: the linear clustering on the set of original data points based on K-means, the nonlinear clustering of each view on the set of linear clusters based on kernel K-means, and multi-view clustering by integrating information from different views for the set of partitions in terms of K-means, which is realized by layers of linear clustering, nonlinear clustering and multi-view clustering in Fig. 1 of this submitted manuscript. It is able to balance the clustering quality, computation cost, completeness and consistency of nonlinear multi-view clustering. For the necessary notations for multi-view data, Wv should come from the linear clustering in Eq. (8) and we wrongly wrote that Wv comes from Eq. (7). According to the physical meaning of different notations (WvUˆv, Wv and Uˆv) and their dimensions, we combine Wv and Uˆv together in Eq (10), where Wv comes from Eq. (8), while Uˆv comes from Eq. (9).
>
> Q2: Following the above point, the motivation of the proposed approach also needs further justification. For example, Eq. (8) applies K-means on the subspace representation. Why one needs to apply K-means on the subspace representation instead of directly applying subspace clustering techniques?
>
> A2: The motivation of the proposed approach can be justified from the following aspects. The motivation of using K-means on the subspace representation as in Eq. (8) is that it corresponds to achieving the goal of efficiency by the linear clustering in the three-level optimization problem. The linear clustering makes data points similar with each other in the local geometric space into the same clusters, which reduces the size of the data for nonlinear clustering. Then, the efficiency is considered in this manner.
>
> Q3: Also, Eq. (9) then applies kernel K-means on the K-means coefficients of the above approach. What is the motivation behind this step?
>
> A3: The motivation of applying kernel K-means on the K-means coefficients of the above approach in Eq. (9) is that it corresponds to achieving the goal of the effectiveness by the nonlinear clustering in the three-level optimization problem.
>
> Q4: Why do we need to further cluster the coefficients?
>
> A4: The motivation of further clustering the coefficients in Eq. (10) is that it corresponds to achieving the goal of consistency and completeness by multi-view clustering in the three-level optimization problem. The reason why we need to further cluster the coefficients is that the final goal of ours is multi-view clustering and Eq. (10) is adopted to achieve this goal.

---

### Official Review · Reviewer_i3oC · 2025-10-26

**Soundness:** 3
**Presentation:** 3
**Contribution:** 3
**Rating:** 6
**Confidence:** 5

**Summary:**

This paper shows that multi-view clustering can be regarded as a three-level optimization problem. The authors flexibly divide the multi-view clustering into three sub-problems based on $ K $-means or kernel $ K $-means, i.e., the linear clustering on the set of original data points, the nonlinear clustering of each view on the set of linear clusters, and multi-view clustering by incorporating partition matrices from different views obtained by linear and nonlinear clustering. The proposed EKNN unifies three different sub-problems into a framework and the authors use the iterative algorithm to optimize the formulated problem. EKNN is able to effectively and rapidly cluster data points from different views. The authors also extend EKNN for multi-view clustering to the case of multi-view subspace learning, which is able to learn a satisfied latent representation shared by different views.

**Strengths:**

The authors flexibly divide the multi-view clustering into three sub-problems based on $ K $-means or kernel $ K $-means, i.e., the linear clustering on the set of original data points, the nonlinear clustering of each view on the set of linear clusters, and multi-view clustering by incorporating partition matrices from different views obtained by linear and nonlinear clustering. The proposed EKNN unifies three different sub-problems into a framework and the authors use the iterative algorithm to optimize the formulated problem. EKNN is able to effectively and rapidly cluster data points from different views.

**Weaknesses:**

1) Overall, the claim of balancing the effectiveness, efficiency, completeness, and consistency of the approach is not fully supported by the current presentation. Likewise for explainability. The effect of each layer in a classical deep neural network is also explainable, i.e., doing convolution plus nonlinear function. The composition of many layers makes the entire network lake of explainability. The proposed approach of combing several clustering steps seems to suffer from the same issue.

2) There is no constraint on the coefficient $\Theta_v$ in Eq. (8). So there exists a trivial solution of $\Theta_v = I$.

3) What is the kernel used in Eq. (9)?

4) There are many typos can be corrected in the whole paper.

**Questions:**

Overall, the claim of balancing the effectiveness, efficiency, completeness, and consistency of the approach is not fully supported by the current presentation. Likewise for explainability. The effect of each layer in a classical deep neural network is also explainable, i.e., doing convolution plus nonlinear function. The composition of many layers makes the entire network lake of explainability. The proposed approach of combing several clustering steps seems to suffer from the same issue.

---

> ### Author Response · Authors · 2025-11-17
>
> Q1: Overall, the claim of balancing the effectiveness, efficiency, completeness, and consistency of the approach is not fully supported by the current presentation. Likewise for explainability. The effect of each layer in a classical deep neural network is also explainable, i.e., doing convolution plus nonlinear function. The composition of many layers makes the entire network lake of explainability. The proposed approach of combing several clustering steps seems to suffer from the same issue.
>
> A1: Thanks for the comment. As mentioned above, the goal of simultaneously considering efficiency, effectiveness, consistency and completeness is achieved by the layer of linear clustering, nonlinear clustering and multi-view clustering in the framework of EKNN, respectively. From the perspective, the claim of balancing the effectiveness, efficiency, completeness, and consistency of the approach is supported. As reviewer mentioned, the effect of each layer in a classical deep neural network is also explainable, i.e., doing convolution plus nonlinear function. The composition of many layers makes the entire network lake of explainability. Regarding the explainability, we then give explanation of the proposed method from the perspective of composition. As convolutional operation in Convolutional Neural Networks (CNN), K-means plays the same role in K-means Neural networks. The composition of many layers based on K-means still makes the entire network owns explainability, i.e., the layer of linear clustering is adopted to achieve the goal of efficiency, the layer of nonlinear clustering is adopted to achieve the goal of effectiveness, the layer of multi-view clustering is adopted to achieve goal of completeness and consistency.
>
> Q2: There is no constraint on the coefficient Θv in Eq. (8). So there exists a trivial solution of Θv=I.
>
> A2: To avoid the trivial solution of Θv = I, there should be a constraint (diag(Θv)=0) on the coefficient Θv in Eq. (8). Considering the case that diag(Θv)=0 can be removed based on the analysis in [a], we do not impose the diag(Θv)=0 constraint to Eq. (8).
> [a] Can-Yi Lu, Hai Min, Zhong-Qiu Zhao, Lin Zhu, De-Shuang Huang, Shuicheng Yan. Robust and Efficient Subspace Segmentation via Least Squares Regression. ECCV (7) 2012: 347-360.
>
> Q3: What is the kernel used in Eq. (9)?
>
> A3: The kernel used in Eq. (9) is Gaussian kernel. We have added this explanation in the place after Eq. (9).
>
> Q4: There are many typos can be corrected in the whole paper.
>
> A4: Thanks for the comment! We will check the whole paper to avoid the possible existing typos.

---

> > ### Comment · Reviewer_i3oC · 2025-11-25
> >
> > The authors have addressed most of my concerns, so I've decided to keep the score.

---

### Official Review · Reviewer_S2aq · 2025-10-27

**Soundness:** 3
**Presentation:** 3
**Contribution:** 3
**Rating:** 8
**Confidence:** 5

**Summary:**

This paper proposes Explainable $ K $-means Neural Networks (EKNN) and present how to unify these three sub-problems into a framework based on EKNN. It is able to simultaneously consider the effectiveness, efficiency, completeness and consistency for the nonlinearly multi-view clustering and can be optimized by an iterative algorithm. EKNN is explainable since the effect of each layer is known. To the best of our knowledge, this is the first attempt to balance the effectiveness, efficiency, completeness and consistency by dividing the multi-view clustering into three different sub-problems.

**Strengths:**

EKNN is explainable since the effect of each layer is known. To the best of our knowledge, this is the first attempt to balance the effectiveness, efficiency, completeness and consistency by dividing the multi-view clustering into three different sub-problems.

**Weaknesses:**

1. One of the main motivations is on efficiency, yet how the proposed method could advance in this regard can be further illustrated.

2. The reason why existing methods in improving efficiency do not suffice is not very clear. In particular, every two algorithms differs in some sense, thus integrating one into the other would increase complexity.

3. Moreover, to compute a n × n matrix Θ directly differs from the motivation that some clustering methods need to compute pairwise similarity between points.

4. Lastly, Eq. (7) and Eq. (8) are very different: Eq. (7) is essentially k-means on the v-th view Xv of data, whereas (8) is doing k-means on the self-representation matrix. Why would the self representation Θ give clusters V that is non-linearly separable as expected ?

**Questions:**

One of the main motivations is on efficiency, yet how the proposed method could advance in this regard can be further illustrated.

---

> ### Author Response · Authors · 2025-11-17
>
> Q1: One of the main motivations is on efficiency, yet how the proposed method could advance in this regard can be further illustrated.
>
> A1: Thanks for these important comments. We have corrected the mentioned issues as follows and checked the whole paper. It includes more clear presentation how the proposed method could advance in efficiency, i.e., we propose Explainable K-means Neural Networks (EKNN) based on K-means, which flexibly integrates the linear clustering, nonlinear clustering and multi-view clustering into a framework. It is able to balance the clustering quality, computation cost, completeness and consistency of nonlinear multi-view clustering. To be specific, the efficiency of the proposed method is realized by the layer of linear clustering in Figure 1. The linear clustering makes data points similar with each other in the local geometric space into the same clusters, which reduces the size of the data for nonlinear clustering. For the layer of nonlinear clustering, we adopt the kernel to map the original data space to high-dimensional space, which usually needs less time cost. Then the efficiency is considered from these perspectives.
>
> Q2: The reason why existing methods in improving efficiency do not suffice is not very clear. In particular, every two algorithms differ in some sense, thus integrating one into the other would increase complexity.
>
> A2: Thanks for the comment. As reviewer pointed, the reason why existing methods in improving efficiency do not suffice is completely unclear in the last submitted manuscript. In particular, every two algorithms differ in some sense, thus integrating one into the other would increase complexity. We aim to show that most existing methods pay more attention to the effectiveness and the improvements in efficiency is not so significant as effectiveness in the last submitted manuscript.
>
> Q3: Moreover, to compute a n × n matrix Θ directly differs from the motivation that some clustering methods need to compute pairwise similarity between points.
>
> A3: The motivation that some clustering methods need to compute pairwise similarity between points is originated from the perspective of the final unified learning framework. The computation of the matrix Θv with n × n dimension is the pre-processing step, which is used as the input to the final unified learning framework. Likewise, the original dataset X is also used as the input. Then directly computing a Θv with n × n dimension does not contradict with the motivation that some clustering methods need to compute pairwise similarity between points from the perspective of the final unified learning framework.
>
>
> Q4: Lastly, Eq. (7) and Eq. (8) are very different: Eq. (7) is essentially k-means on the v-th view Xv of data, whereas (8) is doing k-means on the self-representation matrix. Why would the self representation Θ give clusters V that is non-linearly separable as expected ?
>
> A4: The computation of the matrix Θv with n×n dimension is a pre-processing step, which is used as the input to the final unified learning framework. The obtained self-representation Θv is the extracted feature representation from the original data Xv for the v-th view, which is a refinement of the original Xv. Since the global or local space is built on the relative relationship between data points in the extracted feature representation of Xv for the v-th view, a linearly separable cluster is made up of the refined data points in the dataset and several linearly separable clusters make up a nonlinearly separable cluster. Then the self representation Θv gives clusters V that is non-linearly separable as expected.

---

### Official Review · Reviewer_nCLJ · 2025-10-29

**Soundness:** 3
**Presentation:** 3
**Contribution:** 3
**Rating:** 6
**Confidence:** 5

**Summary:**

This paper proposes Explainable $ K $-means Neural Networks (EKNN) based on $K$-means for multi-view clustering and use the iterative method in $K$-means to optimize the networks. Besides, it is explainable since the effect of each layer in EKNN is knowable, i.e., the layer of kernel $K$-means is adopted to obtain nonlinear clusters. We can also observe that $K$-means and kernel $K$-means are special cases of EKNN. The authors also extend EKNN for multi-view clustering to the case of multi-view subspace learning, which is able to learn a desired latent representation shared by different views. The shared subspace representation can be obtained by the latent representation based on self-expressiveness, which is used to achieve the final results by the existing clustering algorithms, i.e., spectral clustering algorithm.

**Strengths:**

1. This paper proposes Explainable $K$-means Neural Networks (EKNN) based on $K$-means for multi-view clustering and use the iterative method in $K$-means to optimize the networks.
2. Besides, it is explainable since the effect of each layer in EKNN is knowable, i.e., the layer of kernel $K$-means is adopted to obtain nonlinear clusters. We can also observe that $ K $-means and kernel $ K $-means are special cases of EKNN.

**Weaknesses:**

1. Note that in the case when the non-linear feature map from the original data space to the high-dimensional space is explicit, one only need to compute n × k  point cluster distance as opposed to n × n kernel matrix, thus the proposed method is not useful in this case. Hence, an experiment where the features can not be computed and one need to compute the kernel matrix instead is needed. However, the authors did not specify which kernel is used for experiments.
2. The clarity of this paper can be improved. The title and the abstract mentioned this method as Explainable $K$-means Neural Networks. However, there is no usage of neural network in the method.
3. The authors mention in the paper that “the quality of nonlinear multi-view clustering results is guaranteed”. Therefore, more theory can be provided to support this claim.
4. The authors are expected to care for the typo error and check the whole paper to avoid such issues in the whole paper.

**Questions:**

The clarity of this paper can be improved. The title and the abstract mentioned this method as Explainable K-means Neural Networks. However, there is no usage of neural network in the method.

---

> ### Author Response · Authors · 2025-11-17
>
> Q1: Note that in the case when the non-linear feature map from the original data space to the high-dimensional space is explicit, one only need to compute n × k point cluster distance as opposed to n × n kernel matrix, thus the proposed method is not useful in this case. Hence, an experiment where the features can not be computed and one need to compute the kernel matrix instead is needed. However, the authors did not specify which kernel is used for experiments.
>
> A1: Thanks for the comment. As reviewer mentioned, it is important to give the real scenario where computing the kernel matrix is needed, i.e., without considering the real scenario, one only need to compute N × k point-cluster distance as opposed to n × n kernel matrix in the case when the non-linear feature map from the original data space to the high-dimensional space is explicit. In this work, we focus on the real scenario where running time is strictly considered. Since computing the kernel matrix usually needs less time cost, we adopt the kernel to map the original data space to high-dimensional space. We perform experiments to analyze the clustering speed where one computes the kernel matrix, which is shown in the Running Time subsection. The kernel used for experiments is Gaussian kernel.
>
> Q2: The clarity of this paper can be improved. The title and the abstract mentioned this method as Explainable K-means Neural Networks. However, there is no usage of neural network in the method.
>
> A2: Thanks for the comment. As reviewer mentioned, the title and the abstract mentioned this method as Explainable K-means Neural Networks. The reason can be explained from the following aspect. As convolutional operation in Convolutional Neural Networks (CNN), K-means plays the same role in K-means Neural networks. The connections of different K-means components in Figure 1 have the same usage as neural network in CNN. Therefore, we propose Explainable K-means Neural Networks (EKNN) based on K-means for nonlinear multi-view clustering. That is, we extend the definition of neural networks in this manner, which is not limited to CNN.
>
> Q3: The authors mention in the paper that “the quality of nonlinear multi-view clustering results is guaranteed”. Therefore, more theory can be provided to support this claim.
>
> A3: Thanks for the comment. The sentence “the quality of nonlinear multi-view clustering results is guaranteed” is used to explain what is “effectiveness” in this work, which is presented as “effectiveness - the quality of nonlinear multi-view clustering results is guaranteed ” in the last submitted manuscript. The quality of nonlinear multi-view clustering results is the goal what we aim to achieve and the kernel is adopted considering its ability in solving nonlinear cases in this work.
>
> Q4: The authors are expected to care for the typo error and check the whole paper to avoid such issues in the whole paper.
>
> A4: Thanks for the comment. We will check the whole paper to avoid the typo error issues in the whole paper as reviewer suggested.

---

> > ### Comment · Reviewer_nCLJ · 2025-11-25
> > **Comment on 732**
> >
> > The authors have well addressed my concerns and I mantain my original scores.

---

### Comment · Area_Chair_u6eE · 2025-11-24
**Response to Author's Rebuttal**

Dear reviewers,

       The authors now have given their response to the reviews, please have a look on the rebuttal and revised PDF to show your further concerns.

      After that, please give your final rating on this submission.

Your AC
Best

---

### Meta-Review · Area_Chair_f9wn · 2026-01-07

**Summary:**

This paper proposes an explainable k-means neural networks for multi-view clustering. The method is flexible and can effectively and rapidly cluster data points from different views.The reviewers have reached the consensus of acceptance. I also agree with them and tend to accept this paper.

**Reviewer Concerns:**

The major concerns have been addressed.

**Reviewer Scores:**

All reviewers gave positive scores.

---

### Decision · Program_Chairs · 2026-01-26

Accept (Poster)